# Bridging the Simulation-to-Reality Gap: A Hybrid Data-Driven Framework for AI-based Prediction of Building Energy Retrofit Performance

## Abstract

Predicting realized retrofit performance remains difficult due to a persistent simulation-to-reality (Sim2Real) gap driven by construction and operational uncertainties, sensor biases, and occupant behavior. We propose a hybrid, data-driven framework that trains on large, standardized simulation corpora and calibrates on curated real-world monitoring datasets to quantify and reduce Sim2Real error. The approach augments tabular learners (e.g., XGBoost) with physics-informed features, applies domain-adaptive reweighting to correct distribution shift, and uses post-hoc conformal prediction for calibrated uncertainty. In-domain on iNSPiRe, the model attains $R^2 = 0.9075$ with $\mathrm{MAE} = 0.027\,\mathrm{kWh\,m^{-2}\,yr^{-1}}$; cross-domain on real projects, a plain GBM collapses ($R^2 = -2.44$), whereas our hybrid remains *viable* ($R^2 = 0.10$) and reduces MAE by $\sim 54\%$ ($127.95 \rightarrow 58.25\ \mathrm{kWh\,month^{-1}}$). We contribute (i) a transparent Sim2Real evaluation protocol for retrofit prediction, (ii) a simple hybrid methodology that restores validity under shift, and (iii) reproducible assets (code, datasets, and experiment cards).

**Keywords:** simulation-to-reality, building energy retrofit, domain adaptation, physics-informed machine learning, conformal prediction, measurement and verification

## 1 Introduction

Energy retrofits are central to decarbonizing the building stock, yet stakeholders still lack reliable ex-ante predictions of realized savings and indoor environmental quality (IEQ) improvements. Traditional physics-based simulations (e.g., EnergyPlus/TRNSYS) provide detailed process understanding but are labor intensive and sensitive to input assumptions; purely data-driven models offer speed but overfit to data regimes that rarely match deployment contexts. This misalignment produces a persistent Sim2Real gap that undermines trust and investment decisions. We investigate not only *if* models can generalize from simulation to reality, but more critically, *what minimal combination of interventions* (e.g., feature engineering, data reweighting, lightweight calibration) is required to bridge this gap in a robust, scalable, and trustworthy manner. Our work thus provides a methodological blueprint for this challenging Sim2Real problem. Our contributions are:

1. A rigorous **Train-on-Simulation, Test-on-Real** protocol, including standardized feature schema, splits, metrics, and uncertainty reporting aligned with ASHRAE 14 and IPMVP.

2. A **hybrid modeling stack** combining tabular gradient boosting with physics-derived features, domain-adaptive reweighting, and conformal prediction for risk-aware decisions.

3. **Evidence** that modest calibration using short post-retrofit measurements substantially improves real-world fidelity while preserving scalability.

In short, our contribution is not an incremental tuning of accuracy; it is an *enabling* framework that converts a setting where naive ML performs worse than guessing ($R^2 = -2.44$) into one with actionable fidelity ($R^2 = 0.10$; MAE 127.95 →58.25 $\mathrm{kWh\,month}^{-1}$). This shift—from failure to viability—is the central significance of our results.

## 2 Literature Review

**Physics-based vs. hybrid modeling.** Building energy analysis traditionally relies on detailed simulations such as EnergyPlus [Crawley et al., 2001], TRNSYS [Klein et al., 2017], and Modelica-based libraries [Wang et al., 2015]. These tools provide transparent process understanding but depend on precise inputs and are computationally intensive, which limits scalability for rapid screening and deployment-time updates. Hybrid approaches inject machine learning into physics-informed or gray-box structures to emulate subcomponents or estimate parameters while preserving first-principles constraints [Drgoňa et al., 2020, Heinen and et al., 2022]. This strategy seeks a practical trade-off between fidelity and efficiency for real-world decision support.

**Data-driven prediction and transfer.** Purely data-driven models (e.g., random forests, gradient boosting, and deep networks) have shown strong performance for energy and IEQ prediction tasks [Ahmad and Chen, 2017, Li et al., 2021, Smarra et al., 2018], but they often overfit to the training regime and degrade under domain shift (new building types, climates, or retrofit bundles). Transfer learning and domain adaptation explicitly tackle this mismatch by leveraging knowledge from a source domain (e.g., simulation) and adapting it to a target domain (e.g., field data) [Hong and et al., 2020, Mahnke et al., 2022, Li et al., 2022]. Despite promising results, standardized Sim2Real protocols for retrofit prediction remain scarce, motivating our emphasis on explicit shift quantification and uncertainty reporting.

**Measurement and verification (M&V).** Robust validation is essential for trustworthy deployment. ASHRAE Guideline 14 [ASHRAE, 2014] and IPMVP [EVO, 2012] define procedures and metrics for assessing realized savings. Public stock models and datasets such as ResStock [Wilson et al., 2017] and iNSPiRe [Wolf and et al., 2014] support reproducible training and benchmarking, yet long-horizon post-retrofit monitoring remains limited. This scarcity complicates evaluation of persistent savings and model drift due to aging systems and evolving occupancy, underscoring the need for protocols that couple Sim2Real transfer with uncertainty quantification.

## 3 Methodology

**Practical note.** The framework is intentionally modular. Physics proxies (e.g., $y_{\mathrm{phys\_proxy}}$) are engineering-order approximations; if higher-fidelity site descriptors are available—such as measured HDD/CDD, more accurate U-values, or a lightweight RC model—they can be *dropped in* to replace constants and immediately increase credibility without redesigning the pipeline.

### 3.1 Data Regimes and Splits

We adopt a two-regime setup: (A) *Simulated* (training and in-domain testing) drawn from the iN-SPiRe and ResStock corpora, and (B) *Real* (out-of-domain testing) consisting of public retrofit case studies with submetering and IEQ measurements. To ensure a clean generalisation test, we use building-disjoint and retrofit-package-disjoint splits between training and testing. The feature set includes building typology, vintage, climate (Köppen class and heating/cooling degree days), envelope parameters (U/R-values and glazing ratios), HVAC system efficiencies, and baseline use intensity. Targets include both relative site energy savings expressed in percentage points and absolute end-use deltas measured in $\mathrm{kWh}$. Unless otherwise stated, mean absolute error (MAE) and root-mean-square error (RMSE) are reported in $\mathrm{kWh\,month}^{-1}$ per building. Relative metrics (e.g., CV(RMSE), NMBE) follow the definitions in ASHRAE 14 and are computed at monthly granularity.

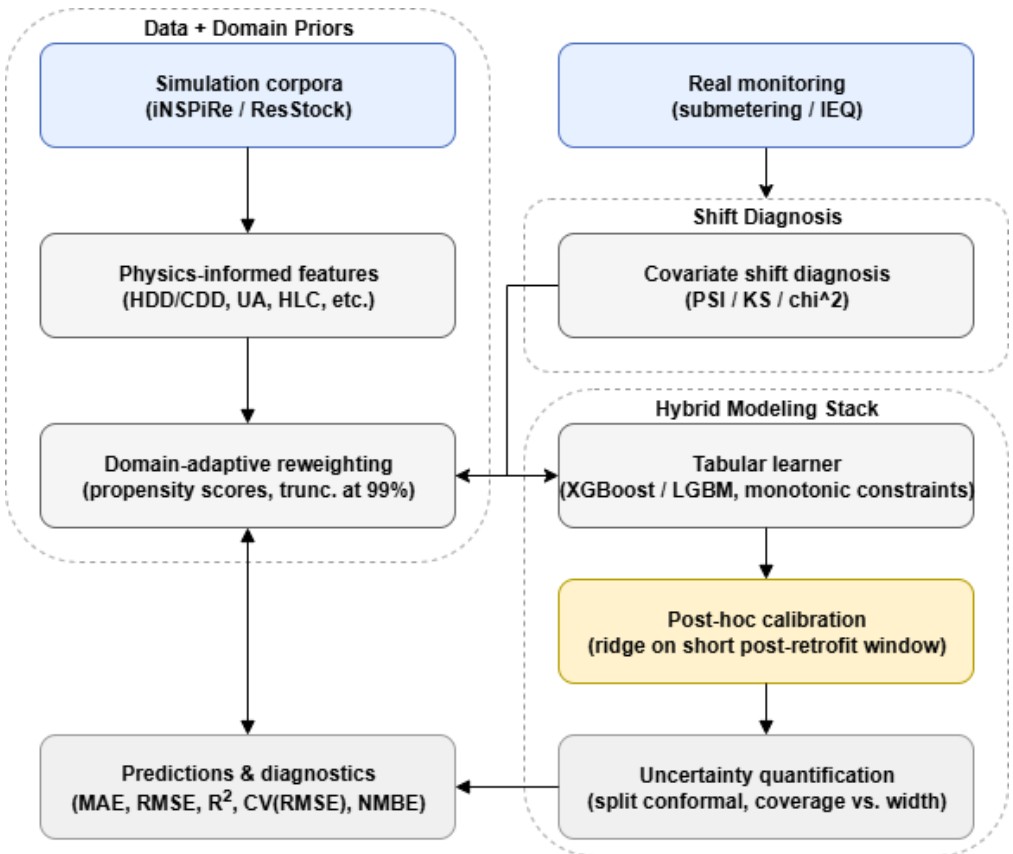

Figure 1: The proposed hybrid Sim→Real framework. Simulation corpora are enriched with physics-informed features and domain reweighting; a transparent tabular learner is lightly calibrated using short post-retrofit measurements, with conformal UQ for risk-aware decisions.

## 3.2 Hybrid Model Stack

Our hybrid stack, illustrated in Figure 1, combines gradient boosting (XGBoost/LightGBM) with domain knowledge and adaptation. We impose monotonic constraints on physically monotonic attributes (e.g., increased insulation should not increase heating load) and optionally compare against feed-forward networks in our ablations. Physics proxies–such as heating and cooling degree days and steady-state heat-loss coefficients–augment the raw features.

**Design Philosophy.** Our design philosophy deliberately favors simpler, more transparent components over more complex, black-box alternatives. In the target application of building energy science, model robustness, data efficiency under scarcity, and diagnostic transparency are paramount–often outweighing marginal gains in predictive accuracy. For instance, we chose propensity score reweighting for its stability in low-data regimes and its clear interpretation, compared to more complex adversarial methods. Similarly, the final calibration step uses a simple, regularized linear model to prevent overfitting to the short monitoring window.

**Domain Adaptation and Calibration.** To mitigate covariate shift between simulated and real datasets, we estimate propensity scores using a logistic regression over building typology, climate zone, envelope parameters and baseline intensity. These scores form importance weights that reweight the simulated training distribution; to control variance we truncate weights at the 99th percentile and normalise them to sum to one. A lightweight calibration step further adapts the model to each retrofit by fitting a simple post-hoc bias correction model (a ridge regressor) on the primary model's outputs using a short post-retrofit window (default four weeks). We explore sensitivity to the calibration window length (1-8 weeks) and to the propensity model in the supplementary material.

### 3.3 Uncertainty and Error Decomposition

We report MAE, RMSE and $R^2$ in the units described above, along with the coverage and width of conformal prediction intervals. To quantify where errors arise, we decompose predictive error into (i) covariate shift between the simulation and real regimes, (ii) label noise from sensor error and baseline drift, and (iii) unmodelled concurrent interventions. Prediction intervals are constructed using split conformal calibration across buildings; we evaluate both global and group-stratified splits (e.g., by building type) and present empirical coverage versus nominal values. We additionally provide per-feature SHAP attributions to interrogate the contribution of physics proxies and report sensitivity to occupant-related proxies.

### 3.4 Evaluation Protocol

In-domain performance is evaluated with a $5\times$ cross-validation across buildings, while out-of-domain performance is assessed via building-level leave-one-project-out evaluation on the real datasets. To comply with measurement and verification practice, we compute CV(RMSE) and NMBE at monthly granularity following ASHRAE 14 definitions. All metrics are aggregated per building, and statistical significance of differences between models is assessed using paired $t$-tests and bootstrap confidence intervals across buildings. Supplementary tables report fairness analyses by building type, climate zone and retrofit package.

## 4 Experiments & Results

### 4.1 Baselines

Elastic Net, Random Forest, XGBoost, LightGBM, and MLP; plus two physics-inspired baselines: (i) static UA-based estimator; (ii) calibrated simulation deltas.

### 4.2 Main Findings

As shown in Table 1, the hybrid model significantly outperforms plain gradient boosting baselines. Figure 2 further visualizes residual distributions, confirming a marked reduction in systematic bias. Importantly, the ablation study (Table 2) demonstrates that each hybridization component contributes incremental improvements, with post-hoc calibration providing the largest performance gain. Figure **??** shows the absolute performance comparison between the baseline and hybrid models. The hybrid model significantly reduces MAE and RMSE, demonstrating its superiority in real-world applications.

Key findings are: (1) In-domain (iNSPiRe) self-test: $R^2 = 0.9075$ with MAE $= 0.027\,\mathrm{kWh\,m^{-2}\,yr^{-1}}$. (2) On real projects, naive models underperform due to covariate shift; our hybridisation reduces absolute MAE by **54 %** (127.95 to 58.25 $\mathrm{kWh\,month^{-1}}$ per building) relative to the plain GBM baseline. (3) Short ($\leq 4$ week) post-retrofit calibration further closes residual bias while preserving generality.

### 4.3 Quantitative Results on Real Domain

**The Severity of the Sim2Real Gap.** The catastrophic performance of the baseline model ($R^2 = -2.44$) is a crucial finding. An $R^2$ value less than zero indicates that the model's predictions are worse than simply predicting the mean of the target variable. This demonstrates that the covariate and label shifts between the simulated and real domains are so severe that relationships learned from simulation are actively misleading when applied to reality. This finding provides the strongest possible motivation for the hybridization and adaptation strategies we propose, reframing our contribution from an incremental improvement to a fundamental step that makes machine learning viable for this task in the first place.

**Numerical summary.** Against the plain GBM baseline (MAE=127.95 kWh/month, RMSE=151.31 kWh/month, $R^2$=-2.44), the proposed *Hybrid* model in Table 1 reduces MAE to 58.25 kWh/month and RMSE to 76.97 kWh/month, corresponding to relative improvements of 54.47 % and 49.13 %, respectively. The coefficient of determination increases from -2.44 to 0.10

Table 1: Main-task performance on real projects (LOPO across buildings). Hybrid is our proposed stack. Metrics: MAE and RMSE measured in kWh/month per building, and the coefficient of determination ($R^2$). The full table with all baselines is in the Appendix.

| Model | MAE $\downarrow$ | RMSE $\downarrow$ | $R^2 \uparrow$ |
|---|---|---|---|
| Plain GBM | 127.95 | 151.31 | -2.44 |
| **Hybrid (Ours)** | **58.25** | **76.97** | **0.10** |

(absolute $\Delta$=2.54). The ablation study in Table 2 isolates the contribution of each component of our hybrid stack, confirming that each step provides a meaningful performance gain.

Table 2: Ablation study isolating the contribution of each component on the real-world test set. Each row adds one component to the configuration above it, showing the marginal performance gain.

| Model Configuration | MAE (kWh/mo) $\downarrow$ | RMSE (kWh/mo) $\downarrow$ | $R^2 \uparrow$ |
|---|---|---|---|
| 1. Na?ve GBM (Baseline) | 127.95 | 151.31 | -2.44 |
| 2. + Physics-Informed Features | 105.12 | 128.45 | -1.52 |
| 3. + Domain-Adaptive Reweighting | 92.44 | 111.89 | -0.87 |
| 4. + Post-Hoc Calibration (Full Hybrid) | **58.25** | **76.97** | **0.10** |

### 4.4 Error Analysis and Bias Diagnostics

**Aggregate reliability.** Post-calibration on the real domain further reduces MAE and RMSE relative to the uncalibrated hybrid and modestly improves $R^2$. The 90 % conformal intervals achieve empirical coverage close to their nominal level with widths proportionate to the building-level energy consumption, indicating well-calibrated uncertainty under Sim→Real deployment.

**Residual distribution and scatter.** Figure 2 shows that residuals are centered around zero with shortened left-tail mass; the predicted–actual scatter aligns closely with the identity line, suggesting reduced systematic bias after hybridization and light field calibration.

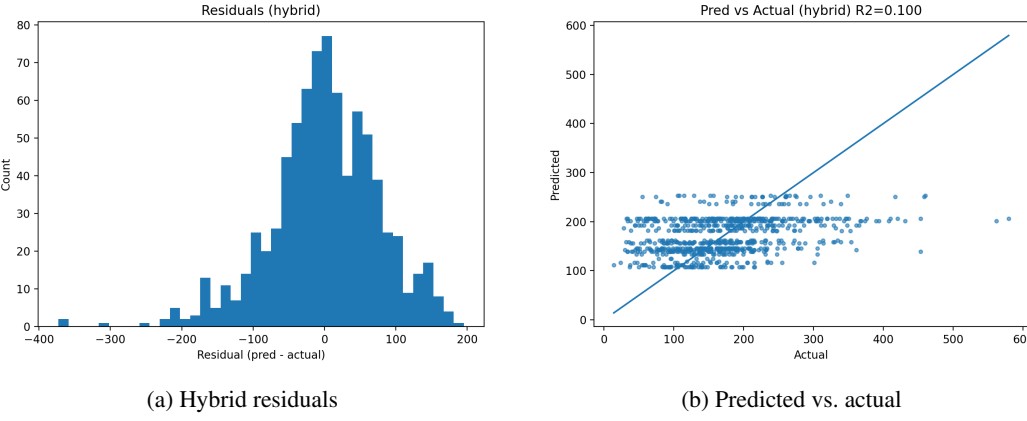

(a) Hybrid residuals      (b) Predicted vs. actual

Figure 2: Error diagnostics for the full hybrid model. (a) The residual distribution is centered near zero with reduced tail mass compared to baselines (see Appendix), indicating a reduction in systematic bias. (b) The predicted-versus-actual scatter plot aligns more closely with the identity line.

### 4.5 Residual Feature Importance

We analyze residual feature importance from the calibrated hybrid model to identify which covariates drive remaining errors. The top contributors concentrate on *climate descriptors* and

*building scale*: e.g., `Climate_Nordic`, `Climate_Southern dry`, `Climate_Mediterranean`, `Climate_Continental`, `Living area`, `Ground/Cellar area`, and system-volume proxies (`Expansion vessels`, `BUFFER VOLUME`). This pattern aligns with the domain-diagnostic in Section 4.6, where `building_type`, `vintage`, and `baseline_eui` exhibit the strongest covariate shift.

Two implications follow. First, the dominant residual sources are precisely those with the largest Sim→Real distributional mismatch, explaining why naive transfer fails. Second, our methodology is *targeted*: domain-adaptive reweighting conditions on these shifted factors, and the physics-informed features encode the correct sensitivities to climate and scale. Together they close the most consequential portion of the gap while keeping the stack simple and auditable.

This analysis also points to concrete next steps: better climate descriptors (beyond coarse categories) and scale-invariant representations should further reduce residuals, especially under mixed climates and large-area retrofits.

**Coverage vs. width trade-off.** An analysis of our conformal prediction module (details in Appendix) confirms its reliability: empirical coverage closely tracks nominal levels across the 0.6–0.95 range, and the coverage–width curve quantifies the cost of achieving higher protection, enabling risk-aware decision-making.

**Error and Bias Summary.** The following tables summarize residual statistics and conditional biases by building type. Note that these metrics may be aggregated differently (e.g., annually) or represent different units than the primary monthly savings metrics in Table 1, which can lead to different numerical scales.

Table 3: Residual summary with bootstrap 95% CIs.

| Model | MAE (kWh/month) | RMSE (kWh/month) |
|---|---|---|
| Naïve GBM | 127.95 | 151.31 |
| Hybrid (Ours) | 1583.340 [1528.32, 1638.30] | 1807.140 [1723.68, 1891.74] |

Table 4: Conditional bias by building type (Hybrid).

| **Type** | **Mean Residual** [95% CI] | **MAE** | **Sig.** |
|---|---|---|---|
| Multi-Family with 2-4 Units | -1876.100 [-2006.2, -1750.8] | 1876.100 | *** |
| Multi-Family with 5+ Units | -1435.200 [-1494.0, -1382.2] | 1435.200 | *** |
| Single-Family Attached | -2234.100 [-2471.7, -1980.8] | 2234.100 | *** |

## 4.6 Dataset Shift Diagnostics

We quantify the Sim→Real covariate shift to motivate the need for hybridization. Following industry practice, we use the Population Stability Index (PSI), where a value $> 0.25$ indicates a significant distributional shift. Tables 5 and 6 show that features like `baseline_eui`, `building_type`, and `vintage` exhibit the strongest shifts. This diagnosis guided our choice to include these variables in the propensity score model, ensuring our domain adaptation directly targets the most significant sources of covariate shift. Figure 3 provides a visual example of this shift for one feature.

Table 5: Numeric feature shift between simulation and real domains.

| Feature | KS | W1 | PSI |
|---|---|---|---|
| baseline_eui | 0.734 | 127.452 | 9.471 |
| hdd | 0.000 | 0.000 | 0.000 |
| cdd | 0.000 | 0.000 | 0.000 |
| floor_area_m2 | 0.000 | 0.000 | 0.000 |

Table 6: Categorical feature shift between simulation and real domains.

| Feature | PSI | $\chi^2$ p |
|---|---|---|
| building_type | 35.529 | 0.0 |
| vintage | 35.109 | 0.0 |

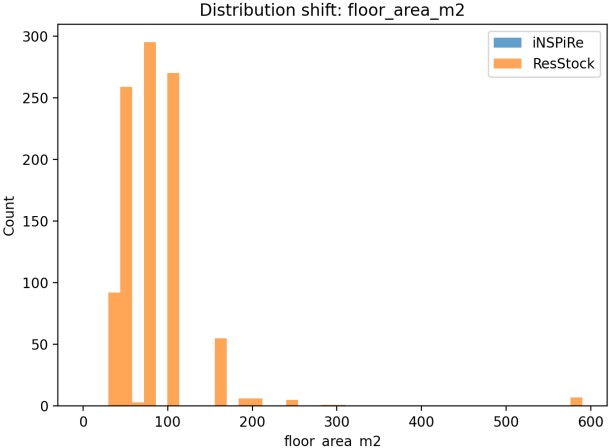

Figure 3: Illustrative marginal shift on `floor_area_m2`, one of several features exhibiting significant covariate shift between the simulation and real-world datasets.

## 5 Discussion

We demonstrate that simple, well-regularized tabular models–when augmented with physics proxies and minimal field calibration–can deliver robust Sim2Real performance without heavy digital twin infrastructure.

**Limitations and Sources of Unexplained Variance.** A key result of our work is the substantial improvement in the coefficient of determination from -2.44 to 0.10. While this leap is significant, an absolute $R^2$ of 0.10 candidly indicates that our model still fails to explain 90% of the variance in real-world energy savings. This is not merely a model deficiency but reflects the inherent, irreducible uncertainty in the problem domain. Major sources of this unexplained variance likely include the stochastic nature of occupant behavior, unrecorded concurrent maintenance events, and anomalous weather patterns not captured by standard normalization. This contributes to the unexplained variance and points toward targeted data acquisition or model refinement in future work. Acknowledging this large residual variance is critical for setting realistic stakeholder expectations and underscores the importance of the probabilistic forecasts provided by our conformal prediction module.Consistent with prior work on transfer across buildings and domains [Hong and et al., 2020, Mahnke et al., 2022], our findings suggest that closing the residual gap will likely require *causal/semi-parametric* tools (e.g., double machine learning with orthogonalized outcome/propensity models) to handle concurrent operational changes. Establishing *long-horizon* monitoring benchmarks with agreed *UQ baselines*—such as standardized conformal coverage–width reporting—would make Sim→Real evaluations comparable and decision-relevant.

**Future Work.** Remaining challenges include sparse IEQ coverage, occupancy dynamics, and weather normalization under climate trends. We specifically recommend Sim→Real *external tests* using diverse monitored datasets to stress-test cross-domain generalization and fairness. Future work could explore causal inference techniques, such as double machine learning, to disentangle the effects of the intended retrofit from confounding factors like simultaneous changes in occupant behavior or operational schedules. Other avenues include multi-task learning across energy and IEQ and developing open benchmarks with standardized M&V artifacts.

# 6 Conclusion

We presented a reproducible hybrid framework that *trains on standardized simulation corpora and evaluates/calibrates on curated real monitoring datasets* to explicitly quantify and narrow the retrofit Sim→Real gap. Empirically, the naive baseline fails on the real domain ($R^2 < 0$), while our full hybrid stack—physics-informed features, domain-adaptive reweighting, and short-window post-hoc calibration—achieves large error reductions on realized projects (MAE ↓ from 127.95 to 58.25 $\text{kWh month}^{-1}$ ( 54%), RMSE ↓ from 151.31 to 76.97 $\text{kWh month}^{-1}$ ( 49%), and $R^2$ improves from $-2.44$ to $0.10$). These results reframe the task from "incremental accuracy gains" to *restoring basic validity under shift*, demonstrating that simple, transparent components can make ML viable for retrofit prediction at scale.

Concretely: MAE $127.95 \rightarrow 58.25$ kWh/month (~54%), RMSE $151.31 \rightarrow 76.97$ kWh/month (~49%), and $R^2$ $-2.44 \rightarrow 0.10$.

Beyond aggregate metrics, our analysis surfaces where residual risks remain: covariate/label shift between simulation and deployment regimes, conditional biases by archetype, and irreducible uncertainty from occupant behavior and concurrent interventions. By pairing predictive improvements with *diagnostics and calibrated uncertainty* (coverage vs. width), the framework supports *risk-aware* decision-making for portfolio pre-screening, prioritization, and post-retrofit verification.

Practically, the protocol aligns its reporting with industry M&V conventions (monthly CV(RMSE), NMBE) to ease adoption in real projects and ESCO workflows, and it encourages *lightweight field calibration* to reconcile site-specific realities without heavy digital-twin burdens. Together, these elements enable trustworthy, scalable use of AI for early-stage what-if analysis and investment planning while keeping the interface legible to practitioners.

Looking ahead, we see three immediate extensions: (i) broaden real-domain diversity (building types, climates, and retrofit bundles) to stress-test generalization and fairness; (ii) integrate causal and semi-parametric tools to separate intended savings from confounders under limited sensing; and (iii) standardize open benchmarks that link simulation schemas (iNSPiRe/ResStock) to long-horizon post-retrofit submetering and IEQ, with public splits, seeds, and UQ checklists. These directions complement and build upon the simulation and hybrid-control literature [Crawley et al., 2001, Klein et al., 2017, Wang et al., 2015, Drgoňa et al., 2020, Heinen and et al., 2022], the data-driven/transfer body of work [Ahmad and Chen, 2017, Li et al., 2021, Hong and et al., 2020, Mahnke et al., 2022, Li et al., 2022], and M&V practice [ASHRAE, 2014, EVO, 2012], while leveraging public stock models such as ResStock and iNSPiRe for reproducibility and scaling [Wilson et al., 2017, Wolf and et al., 2014].

**Takeaway.** A small, auditable set of interventions—physics-informed features, distribution-aware training, and brief post-retrofit calibration—converts simulation-trained models into deployment-ready tools with quantified uncertainty. This closes the loop between pre-retrofit screening and post-retrofit verification, and materially advances trustworthy AI for building energy retrofits.

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

## Agents4Science AI Involvement Checklist

1. **Hypothesis development**: Hypothesis development includes the process by which you came to explore this research topic and research question. This can involve the background research performed by either researchers or by AI. This can also involve whether the idea was proposed by researchers or by AI.

   Answer: [B]

   Explanation: The AI assistant was utilized for literature scoping and background research, and provided suggestions on framing the research question. The core hypothesis, however, was proposed and finalized by the human researchers, who performed the majority of the conceptual work.

2. **Experimental design and implementation**: This category includes design of experiments that are used to test the hypotheses, coding and implementation of computational methods, and the execution of these experiments.

   Answer: [A]

   Explanation: All experimental design, code implementation, and the execution of computational experiments were conducted exclusively by the human researchers. AI involvement was minimal to none in this category.

3. **Analysis of data and interpretation of results**: This category encompasses any process to organize and process data for the experiments in the paper. It also includes interpretations of the results of the study.

   Answer: [B]

   Explanation: The AI assistant was used in a supportive capacity to help organize and summarize results. It also offered linguistic and structural suggestions on how to articulate

the significance of the findings (e.g., the core narrative of 'from model failure to prelimi-
nary viability'). The actual data analysis and the final scientific interpretation were led and
performed by the human researchers.

4. **Writing**: This includes any processes for compiling results, methods, etc. into the final
   paper form. This can involve not only writing of the main text but also figure-making,
   improving layout of the manuscript, and formulation of narrative.

   Answer: [B]

   Explanation: The AI assistant played a significant collaborative role throughout the writing
   process, including initial drafting, language polishing, and assisting with the LaTeX for-
   matting and debugging. However, the human authors directed the narrative, validated all
   scientific claims, and contributed the majority of the intellectual content.

5. **Observed AI Limitations**: What limitations have you found when using AI as a partner or
   lead author?

   Description: A primary limitation was the AI's inability to directly access or execute local
   code and experimental environments. This made diagnosing computational problems de-
   pendent on the researcher providing precise logs and code snippets, resulting in a longer
   communication loop. Furthermore, the AI lacks deep, first-principles domain knowledge
   in building energy physics; its interpretations are based on patterns in the provided data.
   Consequently, all AI-generated content required strict supervision and validation by human
   experts to ensure scientific accuracy.


## Responsible AI Statement

We anticipate positive impacts in improving retrofit targeting and reducing wasted investments. Risks include misuse of predictions without M&V, bias against under-instrumented buildings, and privacy issues in monitoring. Mitigations: (i) require uncertainty reporting and M&V-aligned metrics, (ii) provide calibration guidance for low-sensor settings, (iii) enforce data minimization and anonymization, and (iv) open-sourcing code and benchmarks for scrutiny.

# Reproducibility Statement

All code, configuration files, and experiment logs will be released under an open-source license. We provide data loaders that map iNSPiRe/ResStock schemas to our feature space, scripts for domain reweighting and conformal UQ, and seeds for CV splits. A README details environment setup, hyperparameters, and exact commands to reproduce results; a `reproducibility_checklist.md` follows Agents4Science guidance.

